# Effects of seated Tai Chi Yunshou on upper limb function among stroke patients in the subacute phase: A study protocol for a randomized controlled trial

Xinglai Zhang, Chen Chen, Shaohua Chen, Zhenzhen Ma, Hang Gao, Jiayi Ren, Yuanjia Gu, Weijia Gu, Zhenkun Gao, Guo Lu, Jiming Tao*

Rehabilitation of Shuguang Hospital Affiliated to Shanghai University of Traditional Chinese Medicine, Shanghai, China

* taoyecheng@163.com

## Abstract

### Background

Upper limb dysfunction after stroke is one of the common problems. Tai Chi Yunshou exercise and seated Tai Chi exercise have been confirmed that it is beneficial on upper limb function for stroke patients. Seated Tai Chi Yunshou exercise easier and suitable for stroke patients who are unable to stand. The purpose of this experiment is to explore the effects of seated Tai Chi Yunshou exercise on the upper limb function for stroke patients.

### Methods

84 stroke patients with upper limb dysfunction were randomly divided into two groups. The experimental group and the control group consist of 42 patients each. The experimental group will receive seated Tai Chi Yunshou training combined with conventional treatment, while the control group only will receive conventional training. Each training session will be implemented once a day and five times per week in a duration of 4 weeks. Primary and secondary outcomes will be measured at baseline and 4-weeks, 8-week follow-up after randomization. Fugl-Meyer Upper Extremity Assessment (FMA-UE) will be the primary outcome and Modified Trunk Impairment Scale (mTIS),the Wolf Motor Function Test (WMFT), Functional near-infrared spectroscopy (fNIRS), Barthel Index(BI) will be the secondary outcome. Functional near-infrared spectroscopy(fNIRS) will be used to explore the therapeutic mechanism.

### Discussion

According to previous studies, Tai Chi Yunshou exercise and tailored seated Tai Chi are effective treatment for stroke patients. In this study, we will conduct a seated Tai

**Data availability statement:** No datasets were generated or analysed during the current study. All relevant data from this study will be made available upon study completion.

**Funding:** Key Supported Discipline in Rehabilitation Medicine within the Shanghai Health and Health System (2023ZDFC0301).

**Competing interests:** The authors have declared that no competing interests exist.

Chi Yunshou exercise for patients with stroke. We believe that this study may prove the effectiveness of seated Tai Chi Yunshou exercise on upper limb function among stroke patients.

## Trial registration

ClinicalTrials.gov [ChiCTR2400085317](ChiCTR2400085317)

## Introduction

Upper limb dysfunction is a common and particularly challenging impairment experienced by stroke patients [1]. It is caused by factors such as reduced motor function, sensory loss, ataxia, spasticity, and pain, which hinder the independent performance of functional activities that require upper limb involvement, such as eating, dressing, and washing [2]. About 80% of people with acute stroke have upper limb motor impairment, and of those with reduced arm function early after stroke [3]. There are various methods for upper limb rehabilitation, including muscle strength training, sensory training, motor control training, task-oriented training, constraint-induced movement therapy, and robot-assisted upper limb training. These rehabilitation exercises often emphasize repetitive training of specific body parts, while coordinated training between different limbs is relatively rare, and the interaction between different body parts is seldom prioritized. Meanwhile, the integration of cognitive and motor functions receives little attention [4]. However, the trunk and upper limbs often work in synergy during activities, and the underlying brain mechanisms require further study.

Brain functional specialization emphasizes hemispheric processing, enabling the expanded human brain to minimize inter-hemispheric connectivity and allocate specific processing functions, with relatively less cognitive involvement [5,6]. Improvements in inter-hemispheric functional specialization may enhance the efficiency of cognitive processing [7]. In real-life functional activities, the involvement of different limb movements and cognition often requires coordination between the two hemispheres. Tai Chi is a holistic mind-body activity that helps practitioners focus their attention, regulate breathing, and improve motor function [8]. It emphasizes coordination between different limbs, the interaction between movement and cognition, and the coordination between movement and breathing, focusing on the connection between the two cerebral hemispheres. Long-term Tai Chi practice can optimize functional connectivity in brain regions that play a crucial role in cognitive-motor integration [9]. Performing the Tai Chi Yunshou movement leads to higher activation of the motor cortex and prefrontal cortex, which, in the long term, results in enhanced or remodeled functional connectivity in brain regions [6,10].

In the present study, we situate our investigation within established neurorehabilitation frameworks—including Dynamic Systems Theory (DST), Task-Oriented Training (TOT), and the International Classification of Functioning, Disability and Health (ICF)—to theoretically ground the application of seated Tai Chi Yunshou for post-stroke recovery. Guided by DST, we conceptualize Yunshou not merely as

exercise but as a task-specific constraint that promotes self-organization of the neuro-musculo-skeletal system toward more functional and stable coordination patterns, facilitated by its rhythmic, bilateral, and multi-joint nature [11]. Its movement patterns, such as pushing, pulling, and weight-shifting, mirror components of activities of daily living, aligning with TOT principles that emphasize functional, goal-directed practice to promote neuroplasticity [12]. Potential outcomes are framed within the ICF model, spanning improved interhemispheric connectivity (Body Function & Structure), enhanced bimanual coordination and upper limb function (Activity), and increased engagement in daily life (Participation). Current research has demonstrated the benefits of seated Tai Chi and Tai Chi Yunshou for upper limb function in stroke patients [11–13]. Tai Chi is an exercise centered around the waist, and compared to regular Tai Chi, seated Tai Chi emphasizes the linkage between the trunk and upper limbs [14]. Tai Chi is a complex exercise that is difficult to learn in a short period. The Yunshou movement in Tai Chi is a fundamental form that well reflects the essence of Tai Chi [15–17]. However, many subacute stroke patients are unable to perform Tai Chi Yunshou training due to poor standing balance. Therefore, this study aims to investigate the effects of seated Tai Chi Yunshou training on upper limb function and brain functional mechanisms in subacute stroke patients.

## Methods

### Study design

A single-center, parallel, randomized controlled trial was designed to explore the effects of seated Tai Chi Yunshou on upper limb function in patients with subacute stroke. This trial design was approved by the Ethics Committee of Shuguang Hospital Affiliated to Shanghai University of Traditional Chinese Medicine (Approval No. 2023-1414-181-01) and registered on the Chinese Clinical Trial Platform (Registration No. ChiCTR2400085317).

### Participants

All subjects will be recruited from the rehabilitation department of Shuguang Hospital Affiliated to Shanghai University of Traditional Chinese Medicine, with recruitment taking place from September 1, 2025, to September 30, 2026. Subjects will be recruited based on the following criteria:

1. Doctors will screen patients admitted to the hospital and referred those who might meet the inclusion criteria for the study to the research leader.

2. The researchers will introduce the concept of the trial to potential patients and inquire whether they will be willing to participate voluntarily.

3. Subjects will undergo an assessment process.

4. Subjects will sign an informed consent form before the start of the trial.

**Inclusion criteria.** In accordance with the diagnostic criteria of the 'Chinese guidelines for diagnosis and treatment of AIS' (2018 version)and confirmed by CT or MRI examination.
First episode of stroke
Onset time between 2 weeks to 6 months
Seated balance at Level 1 or above
Unilateral limb impairment
Able to tolerate 30 minutes of seated Tai Chi Yunshou exercise and signed informed consent.
**Exclusion criteria.** MMSE score of 23 or less
Unable to tolerate 30 minutes of sitting Tai Chi Yunshou exercise
Presence of other serious illnesses, such as heart disease, kidney failure, tumors, etc.

Patients will be randomly assigned to the experimental group or the control group based on a computer-generated random sequence. Before the trial, patients will receive a sealed, opaque envelope determining their group assignment. A dedicated person will be responsible for data collection and group assignment. Assessments will be conducted by physical therapists who will be unaware of the experimental grouping and treatment plan. Participants in the trial will not be allowed to exchange information.

## Rejection criteria

Participants who breach the predetermined trial protocol at any point during the study period will be disqualified.

If a participant's health condition destabilizes or demonstrates signs of deterioration throughout the course of the study, they will be promptly excluded.

Data sets that deviate from actual findings due to experimental equipment malfunctions or abnormalities will be rejected.

## Interruption criteria

In the event of serious adverse events, if deemed significant by the attending physician, clinical observation for the individual case shall be immediately ceased.

Should the disease condition deteriorate during the clinical trial, or if other disease manifestations emerge that hinder the observation of the case, as assessed by the attending physician, the clinical trial will be halted.

Substantial deviations from the clinical trial protocol, including poor compliance or participants expressing reluctance to continue clinical observation and requesting withdrawal from the study to the supervising physician or investigator, will result in the termination of the subject's involvement.

## Sample size

This trial aims to assess the impact of Tai Chi on upper limb function in post-stroke patients. Using G-power V.3.1.9.2 software for sample size calculation, the Fugl-Meyer Assessment Upper Extremity (FMA-UE) scale was selected as the primary outcome measure. Based on data from a previously published study [18], the mean FMA-UE scores post-intervention were $60.50 \pm 2.19$ for the Tai Chi group and $59.14 \pm 1.77$ for the control group. To detect the targeted effect size of 0.6830377 with a type 1 error rate of 5% ($\alpha = 0.05$) and 80% statistical power ($\beta = 0.20$), a sample size of 64 participants is deemed sufficient. Accounting for a potential dropout rate of 20%, a total of 84 participants will be enrolled, with 42 participants assigned to each group.

## Randomization and blinding

Eligible subjects who fulfill the inclusion criteria and provide signed informed consent will undergo randomization, allocated in a 1:1 ratio to either the control or intervention groups. The randomization process will be computer-generated using a block randomization design with a fixed block size of 4, implemented by independent statistical experts blinded to study details. The random allocation sequence will be produced using SPSS software to ensure randomness and reproducibility. To maintain the integrity of the randomization, therapists not directly involved in recruitment, assessment, or intervention will securely place each allocated number into an opaque envelope. Following patient screening, the recruitment physician will liaise with a designated coordinator who will then dispatch the necessary documentation to eligible subjects, informing them of their group assignment. This rigorous randomization procedure aims to minimize bias and ensure the validity of the study results.

Owing to the inherent characteristics of the intervention, it is not feasible to blind the subjects and therapists directly involved in the intervention process. However, throughout the duration of the study, the therapist conducting assessments, the researcher tasked with data collection, and the statistician analyzing the data will remain blinded to group allocations.

This blinding protocol will be maintained until the completion of statistical analyses to minimize potential bias. Once the statistical analyses are concluded, the blinding will be lifted.

### Intervention

Both groups received conventional rehabilitation therapy, with the rehabilitation plan based on assessment. The regular treatment included muscle strength training, sensory training, functional activity training, seated balance training, acupuncture, biofeedback therapy, and more. The total duration of the therapy was 3 hours per day, 5 days a week, for a total of 4 weeks. The experimental group received an additional 30 minutes of seated Tai Chi Yunshou training, consisting of a 15-minute one-by-one session with an experienced Tai Chi therapist in the morning, followed by a 15-minute seated Tai Chi Yunshou practice in the afternoon under the supervision of a caregiver.

### Control group

The control group solely adhered to a program of conventional comprehensive rehabilitation training which consists of physical therapy, acupuncture, Tuina, speech therapy if necessary. Physical therapy includes rolling, side lying to sitting, sitting training, sitting to standing, walking training. motor control training, muscle strength training, Tuina massage for 60 mins.

The acupuncture protocol involved a standardized selection of points categorized by anatomical region: Scalp and Face: Baihui (GV20), Sishencong (EX-HN1), Tianzhu (BL10), and Lianquan (RN23). Upper Limb: Jianyu (LI15), Quchi (LI11), Waiguan (TE5), and Hegu (LI4). Lower Limb: Huantiao (GB30), Zusanli (ST36), Xuanzhong (GB39), Sanyinjiao (SP6), and Taichong (LR3) [19]. All acupoints were stimulated using a manual high-frequency, small-amplitude twirling technique, administered by licensed practitioners with ≥5 years of clinical experience. Needle retention lasted 20 minutes per session.

To safeguard the well-being of the participants, a one-on-one training modality was implemented between the therapist and subjects. This arrangement ensured that training was discontinued promptly in the event of subject-reported fatigue or any discomfort.

### Experimental group

The intervention program for the test group entailed a combination of seated Tai Chi Yunshou exercises and routine comprehensive rehabilitation training [11]. Under the guidance of the therapist, patients were taught the seated Tai Chi Yunshou exercises.

Specific training methods are as follows: 1. starting position. Sit on a stable chair or bed with both feet flat on the ground, shoulder-width or slightly wider. Let both hands hang naturally at your sides, relax whole body, and look straight ahead. Keep spine naturally straight, without leaning too much forward or backward. 2.Beginning the Movement (Left Yunshou as an Example). Slowly lift left hand from the left side of body, palm facing inward, while simultaneously lifting right hand to chest, palm facing downward. Turn upper body slightly to the left, and as turning, move left hand in an upward arc to the left, gradually flipping palm outward. At the same time, move right hand in a downward arc to the right, palm facing downward. Continue moving left hand in an arc to the left and backward, palm facing outward, while simultaneously moving right hand in an arc from the front of abdomen to the left and upward to left chest, palm facing inward. Follow the movement of left hand with eyes, keeping head straight and avoiding tilting or tilting up and down.3. Alternate Movement. Next, perform the Right Yunshou, with movements opposite to the Left Yunshou. Alternate between the two hands to create a continuous flow of Yunshou movements. Pay attention to maintaining body stability, using the waist as the axis to drive the hands in arcs.

### Measures

In this study, the primary outcome and secondary outcomes will be measured at baseline, the end of the 4-week treatment and the end of the 8-week follow-up. All outcome assessments will be independently performed by experienced and blinded assessors, who are also professional physical therapists.

### Primary outcome measures

**Fugl-Meyer Upper Extremity Assessment (FMA-UE).** The Fugl-Meyer Assessment Scale (FMAS) provides a comprehensive evaluation for stroke patients with hemiplegia, covering motor, sensory, balance, joint mobility, and pain domains. In our study, we focused on the upper extremity segment, which includes 33 assessment items worth 66 points, capturing various motor aspects. This enables precise evaluation which is basic for specific therapeutic interventions for stroke patients.

### Secondary outcome measures

**Modified Trunk Impairment Scale (mTIS).** The mTIS can be used to examine the degree of trunk control or the level of trunk impairment, which is seen as a prerequisite for balance, gait, motor function, and activities of daily living performance in stroke survivors. It comprises two distinct subscales: the Dynamic Sitting Balance Scale (DBS) and the Coordination Scale (CS). The DBS encompasses four categories, specifically assessing elbow contact with the bed on the hemiplegic and healthy sides, as well as pelvic elevation from the bed on both the hemiplegic and healthy sides. Scores for the first two categories range from 0 to 3, while scores for the latter two categories vary from 0 to 2. Meanwhile, the CS focuses on alternating forward rotations of the shoulders and knees, assigning scores from 0 to 3 for each. The total score range of this scale is 0–16, with higher scores indicative of lesser impairment in trunk control, as outlined in previous literature.

**The Wolf Motor Function Test (WMFT).** The Wolf Motor Function Test (WMFT) will be used to evaluate the functional activity of the upper limbs. The WMFT consists of 15 items. Items 1–6 are simple joint movements while items 7–15 are compound functional movements. The WMFT score is determined based on the quality of all actions (0–5 points, six grades).

**Functional near-infrared spectroscopy (fNIRS).** fNIRS employs a continuous-wave optical system to quantify oxygenated hemoglobin (HbO) levels in the cerebral cortex. The system utilizes a light source that emits near-infrared light at wavelengths of 690 nm and 830 nm, operating at a fixed sampling rate of 10 Hz across 95 channels. A 3D digitizer is utilized to acquire precise coordinates for each channel. These coordinates are subsequently transformed into Montreal Neurological Institute (MNI) coordinates, allowing for the projection of the data onto the standard MNI brain model using the NIRS_SPM toolbox. With this configuration, the channels are capable of covering key brain regions including the supplementary motor area (SMA), primary motor cortex (PMC), primary motor cortex (M1), primary sensory cortex (S1-S5), and dorsolateral prefrontal cortex (DLPFC) [10].

We hypothesize that seated Tai Chi Yunshou will elicit increased activation in motor-related regions (M1, SMA) and prefrontal areas (DLPFC) due to its integrated motor-cognitive demands. Raw fNIRS signals will be preprocessed using motion artifact correction (e.g., wavelet-based filtering), bandpass filtering (0.01–0.2 Hz), and conversion to hemoglobin concentrations via the Modified Beer-Lambert Law. To address multiple comparisons, we will apply false discovery rate (FDR) correction for channel-wise analyses and cluster-based permutation testing for spatial inferences. Furthermore, interhemispheric connectivity and functional remapping will be objectively quantified using coherence and correlation-based metrics between homologous channels, with graph theory approaches assessing network efficiency changes. These metrics will be correlated with clinical outcomes to link neural changes to functional recovery.

**Barthel Index(BI).** The participants' activities of daily living (ADL) will be evaluated during the intervention and follow-up periods using the Barthel Index (BI). Previous research has established the BI as a highly reliable tool for assessing ADLs in stroke survivors, demonstrating strong inter-rater and intra-rater reliabilities. The BI comprises 10 items that describe various daily activities, with lower scores indicating a higher level of dependency in performing these tasks.

**Montreal Cognitive Assessment(Moca).** The Montreal Cognitive Assessment (MoCA) is a validated screening tool designed to evaluate cognitive function across six domains: memory, executive function, language, attention, calculation ability, and visuospatial skills. Administered by a trained evaluator, the test comprises a series of tasks and questions, with responses scored against standardized criteria. Total scores range from 0 to 30, where higher scores reflect superior

cognitive performance. A score below 26 points is commonly used as a threshold to identify potential cognitive impairment (CI) or mild cognitive impairment (MCI), though cutoff values may vary based on educational attainment and age.

### Data management

Original study data will be prospectively collected using standardized paper-based case report forms (CRFs) at predefined assessment timepoints. Two independent data managers, both blinded to treatment allocation, will perform dual data entry into EpiData Entry Software (version 3.1) to ensure accuracy. Prior to study initiation, these personnel will complete certified training in Good Clinical Practice (GCP) standards for data management. The Data Monitoring Committee (DMC) will conduct quarterly audits verifying: (1) adherence to data handling procedures; (2) consistency between duplicate entries; and (3) protocol-defined quality control metrics. To protect participant confidentiality, all datasets will undergo full de-identification (removal of direct identifiers, replacement with non-traceable study codes) before statistical analysis. Upon completion of data collection, the finalized database will be locked via restricted access controls to prevent unauthorized modifications. Archived data will be securely stored for a minimum retention period of five years post-study completion, with limited access permitted solely for regulatory inspections or protocol-approved secondary analyses.

### Statistical analysis

Statistical analyses were conducted utilizing IBM SPSS 26.0 software. In terms of descriptive statistics, qualitative indicators were represented by percentages, while quantitative indicators were summarized using means and standard deviations. For quantitative data, an independent samples t-test was applied when the data followed a normal distribution. In cases where normality assumptions were not met, the non-parametric Wilcoxon rank sum test was employed. Qualitative data were analyzed using the chi-square test, and rank data were examined with the Wilcoxon rank sum test. For datasets containing multiple observation time points, a repeated measures ANOVA was utilized to account for the longitudinal nature of the data. Statistical significance was determined at a P-value less than 0.05. For longitudinal data violating normality/sphericity, the non-parametric Friedman test replaced repeated measures ANOVA.

Given the limited sample size (N = 84) in this study, Multiple Imputation (MI) was employed to address missing data. This method generates multiple complete datasets and pools analysis results, thereby mitigating the bias associated with single imputation in small-sample studies. All imputation procedures were conducted using the MICE package in R, with five parallel imputations performed to ensure stability.

### Quality control

To ensure study integrity, comprehensive quality control measures will be implemented throughout the trial. Monitoring data will be systematically submitted to the Steering Committee and archived for longitudinal reference. Comprising Principal Investigators (PIs) from the four participating research centers, this committee will oversee trial coordination, protocol implementation, and quality assurance across all study programs. All research personnel must complete mandatory training covering trial methodology, standardized operating procedures, and monitoring protocols prior to study participation. Post-training competency will be verified through standardized assessments to ensure methodological consistency. Protocol modifications will follow a structured approval process: amendments must be reviewed by the Steering Committee and formally approved by the Institutional Review Board (IRB). All revisions will be documented in a protocol amendment log, detailing the nature of changes, approval dates, and version control identifiers.

## Discussion

This study represents the first clinical trial to evaluate the effects of seated Tai Chi Yunshou exercise on patients with subacute stroke. In this trial, we assessed the impact of seated Tai Chi Yunshou training combined with conventional training compared to conventional training on upper limb function in patients with subacute stroke.

Seated Tai Chi Yunshou exhibits distinct mechanistic advantages compared to conventional rehabilitation approaches. In contrast to bilateral arm training—which often involves symmetric, rhythmic, and at times passive movement patterns—Yunshou emphasizes active and intentional control (known as "Yi" in Tai Chi philosophy), requiring continuous conscious modulation of movement quality, trajectory, and posture. Moreover, Yunshou incorporates asymmetric yet highly coordinated bimanual patterns, such as "separating hands" (Fen Shou), where one hand leads while the other follows, thereby demanding more sophisticated neural control and integration than purely symmetric movements [20]. The seated posture further facilitates integrated trunk–arm coordination through sustained subtle core activation that is dynamically coupled with upper limb movement, rather than promoting isolated limb motion. Unlike therapist-guided interventions such as proprioceptive neuromuscular facilitation (PNF), which relies on specific diagonal patterns and external resistance to evoke neuromuscular responses, Yunshou is self-initiated and depends on internal rhythm and proprioceptive feedback. This quality may enhance the engagement of internal modeling and predictive control mechanisms [21]. Additionally, the continuous and fluid nature of Yunshou, characterized by smooth transitions between movement phases, imposes unique challenges on neural timing and coordination compared to the segmented and discrete structure of PNF. We propose that the integration of continuous mindful control, asymmetric bimanual coordination, and dynamic trunk–limb synergy may uniquely foster neuroplastic adaptations and enhance interhemispheric connectivity following stroke.

Trunk function is pivotal for functional activities, and augmenting trunk performance can expedite the restoration of upper limb functionality [22]. Prior research has attested to the efficacy of seated Tai Chi exercises in enhancing trunk function for patients in the subacute phase of stroke [11]. The seated Tai Chi Yunshou exercises have the potential to improve upper limb function through four primary mechanisms. Firstly, these exercises incorporate bilateral upper limb function training and bilateral upper limb training is beneficial for stroke patients [23]. Secondly, by centering on waist movement, the seated Tai Chi Yunshou exercises engage both the trunk and pelvic [14], with the potential to refine trunk function and consequently enhance upper limb performance. Thirdly, emphasis on breathing during these exercises, particularly deep inhalation accompanying abdominal contraction, activates the diaphragm and transversus abdominis—key core muscles [24]. activating these muscles further bolsters trunk function, ultimately leading to improved upper limb function. Last not the least, as an adapted element of Tai Chi, the seated Tai Chi Yunshou exercises emphasize a comprehensive mind-body movement that incorporates concentration, visual tracking, and engagement of multiple body parts. This holistic approach stimulates interhemispheric connections, fosters the activation of cognitive-motor pathways, and underscores the interplay between cognition and movement, potentially accelerating the recovery of upper limb function [25].

Previous studies have shown that Tai Chi training, tailored seated Tai Chi training, and Tai Chi Yunshou training all have beneficial effects on improving balance and functional activities for stroke patients. However, Tai Chi is a relatively complex exercise that requires a longer period of learning. For many patients with poor functional abilities or mild cognitive impairment, it can be difficult to practice in a clinical setting. Sitting Tai Chi Yunshou training can meet the needs of such patients.

## Limitation

This study has four limitations. Firstly, the sample size of this study is relatively small. Secondly, the study did not evaluate the contraction of trunk muscles when patients performed seated Tai Chi Yunshou exercise. Thirdly, the intervention period was only 4 weeks, which is relatively short. Finally, there was no long-term follow-up, so it is unknown what the long-term effects of seated Tai Chi Yunshou exercise are on stroke patients.

## Trial status

The study registration number is ChiCTR2400085317 and the recruitment still in progress. The recruitment may be finished in September 2026.

## Supporting information

**S1 File. SPIRIT checklist.**
(DOCX)

**S2 Fig. SPIRIT-Figure.**
(DOC)

**S3 File. Research protocol(Chinese).**
(DOC)

**S4 File. Research Protocol (English).**
(DOC)

**S5 Fig. Flow chart of study procedure.**
(PNG)

## Acknowledgments

The authors would like to thank the patients and therapists who cooperated in this study.

## Author contributions

**Conceptualization:** Xinglai Zhang, Jiming Tao.

**Data curation:** Chen Chen, Guo Lu.

**Funding acquisition:** Jiming Tao.

**Investigation:** Shaohua Chen.

**Methodology:** Shaohua Chen, Zhenzhen Ma, Yuanjia Gu, Weijia Gu, Zhenkun Gao.

**Project administration:** Zhenzhen Ma, Yuanjia Gu.

**Resources:** Zhenzhen Ma, Guo Lu.

**Software:** Hang Gao.

**Supervision:** Hang Gao, Yuanjia Gu, Weijia Gu.

**Validation:** Hang Gao, Weijia Gu.

**Visualization:** Jiayi Ren.

**Writing – original draft:** Xinglai Zhang.

**Writing – review & editing:** Xinglai Zhang, Jiayi Ren.

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
