## [Decision Letter · Decision Letter 0]

7 Aug 2025

Dear Dr. Xinglai Zhang,

Thank you for submitting your manuscript to PLOS ONE. After careful consideration, we feel that it has merit but does not fully meet PLOS ONE’s publication criteria as it currently stands. Therefore, we invite you to submit a revised version of the manuscript that addresses the points raised during the review process.

We look forward to receiving your revised manuscript.

Kind regards,

Mansour Abdullah Alshehri

Academic Editor

PLOS ONE

Journal Requirements:

“Key Supported Discipline in Rehabilitation Medicine within the Shanghai Health and Health System�2023ZDFC0301”

“No Conflict of Interest”

Reviewers' comments:

Reviewer's Responses to Questions

**Comments to the Author**

1. Does the manuscript provide a valid rationale for the proposed study, with clearly identified and justified research questions?

Reviewer #1: Yes

Reviewer #2: Yes

2. Is the protocol technically sound and planned in a manner that will lead to a meaningful outcome and allow testing the stated hypotheses?

Reviewer #1: Yes

Reviewer #2: Yes

3. Is the methodology feasible and described in sufficient detail to allow the work to be replicable?

Reviewer #1: Yes

Reviewer #2: Yes

4. Have the authors described where all data underlying the findings will be made available when the study is complete?

Reviewer #1: Yes

Reviewer #2: Yes

5. Is the manuscript presented in an intelligible fashion and written in standard English?

Reviewer #1: Yes

Reviewer #2: Yes

You may also provide optional suggestions and comments to authors that they might find helpful in planning their study.

Reviewer #1: The protocol planned is fairly well thought out and written. However, there are several comments following which should be addressed to improve the statistical design, analysis and data management and quality control.

The sample size design is written reasonably clearly with anticipated dropouts.

The effect size appears to have been calculated with a standard deviation of 2 given the current data in the sample size section. Is that the only source of past data? What about the literature, if any?

The initial patient assessment appears to be ongoing as per the sections on Participants and Exclusion criteria which appear to be written in the past tense. There should be an updated writing as to the exact assessments being made for these initial assessments.

Randomization and blinding are reasonable. Primary and secondary endpoints are presented.

The statistical analysis is expected to be a repeated measures ANOVA with alternative non parametric Wilcoxon rank sum. However, a repeated measures nonparametric alternative should be in place as well, should it be needed. The time line figure looks reasonable.

With such a small sample is imputation as noted in the protocol really reasonable? If so, presumably there will be an imputed and complete data set analysis for quality control purposes and data interpretation checks.

There should be a section on data management and quality control. Section 2.5 in the protocol supplement requires more detail on the actual quality control of the data and data flow.

Planned management of dropout cases is well planned as per the supplemental protocol.

The concerns above should be addressed by the investigators.

Reviewer #2: The study draws on emerging evidence highlighting the importance of motor-cognitive integration and bilateral upper limb synergy in stroke rehabilitation. It integrates functional near-infrared spectroscopy (fNIRS), providing a neurophysiological perspective that aligns well with modern neurorehabilitation paradigms.

The theoretical framework remains descriptively broad and lacks integration with established neurorehabilitation models, such as Dynamic Systems Theory, Task-Oriented Training, or the International Classification of Functioning (ICF). The authors suggest Tai Chi optimizes interhemispheric connectivity, yet do not clearly define mechanisms through which seated Tai Chi Yunshou would differ in this effect from other bilateral or trunk-focused interventions (e.g., bilateral arm training or PNF). The use of “mind-body coordination” and “cognitive-motor synergy” is conceptually promising but needs more rigorous operationalization.

While randomization is computer-generated and opaque envelopes are used, the lack of block randomization or stratification (e.g., based on stroke severity or dominant hand) may compromise balance across groups.

The control group receives conventional therapy, including a broad range of interventions (acupuncture, Tuina, speech therapy), which may vary widely across patients. Without standardized delivery, this introduces treatment heterogeneity, weakening internal validity. The authors do not quantify therapy dose equivalently across groups (i.e., whether total contact time is matched), risking confounding from unequal therapeutic exposure.

The inclusion of fNIRS is forward-looking, but the analysis plan lacks detail. No hypotheses are presented about expected regional activation patterns (e.g., M1, SMA, DLPFC), and no preprocessing or statistical correction strategy (e.g., FDR, Bonferroni) is outlined for handling fNIRS’ multiple comparisons problem.

The manuscript refers repeatedly to “enhanced interhemispheric connectivity” and “functional remapping,” yet no objective functional connectivity analyses (e.g., correlation or coherence matrices from fNIRS) are proposed.

The authors claim this is a mind-body therapy but do not measure cognitive, attentional, or emotional outcomes (e.g., MoCA, Stroop test, or depression/anxiety scales), leaving the “mind” component entirely theoretical.

**Do you want your identity to be public for this peer review?** For information about this choice, including consent withdrawal, please see our Privacy Policy

Reviewer #1: No

Reviewer #2: No

---

## [Author Response · Author response to Decision Letter 1]

21 Sep 2025

Dear Reviewers

We have corrected the manuscript to meet your suggestion. thanks very much for your reminder.

---

## [Decision Letter · Decision Letter 1]

2 Oct 2025

Effects of seated Tai Chi Yunshou on upper limb function among stroke patients in the subacute phase: a study protocol for a randomized controlled trial

PONE-D-25-10868R1

Dear Dr. Zhang,

We’re pleased to inform you that your manuscript has been judged scientifically suitable for publication and will be formally accepted for publication once it meets all outstanding technical requirements.

Kind regards,

Mansour Abdullah Alshehri

Academic Editor

PLOS ONE

Additional Editor Comments (optional):

Reviewers' comments:

Reviewer's Responses to Questions

**Comments to the Author**

1. Does the manuscript provide a valid rationale for the proposed study, with clearly identified and justified research questions?

Reviewer #1: Yes

2. Is the protocol technically sound and planned in a manner that will lead to a meaningful outcome and allow testing the stated hypotheses?

Reviewer #1: Yes

3. Is the methodology feasible and described in sufficient detail to allow the work to be replicable?

Reviewer #1: Yes

4. Have the authors described where all data underlying the findings will be made available when the study is complete?

Reviewer #1: Yes

5. Is the manuscript presented in an intelligible fashion and written in standard English?

Reviewer #1: Yes

You may also provide optional suggestions and comments to authors that they might find helpful in planning their study.

Reviewer #1: All my issues have been addressed with incorporations into the final revised protocol.

XXXXXXXXXXXXXXXXXXXXXXXXXXXXXXXXXXXXXXXXXXXXXXXXXXXXXXXXXXXXXXXXXXXXXXXXXXXXXXXXXXXXXXXXXXXXXXXXXXXXXXXXXXXXXXXXXXXXXXXXXXXXXXXXXXXX

**Do you want your identity to be public for this peer review?** For information about this choice, including consent withdrawal, please see our Privacy Policy

Reviewer #1: No

---

## [Editor Report · Acceptance letter]

PONE-D-25-10868R1

PLOS ONE

Dear Dr. Zhang,

I'm pleased to inform you that your manuscript has been deemed suitable for publication in PLOS ONE. Congratulations! Your manuscript is now being handed over to our production team.

Kind regards,

on behalf of

Dr. Mansour Abdullah Alshehri

Academic Editor

PLOS ONE